# Margin Status Post Cervical Conization Predicts Residual Adenocarcinoma In Situ (AIS) and Occult Adenocarcinoma in a Predominantly Hispanic Population

**DOI:** 10.3390/diagnostics11101889

**Published:** 2021-10-13

**Authors:** Linda J. Hong, Sandy Huynh, Joy Kim, Laura Denham, Mazdak Momeni, Yevgeniya J. M. Ioffe

**Affiliations:** 1Department of Obstetrics & Gynecology, Loma Linda University School of Medicine, Loma Linda, CA 92354, USA; lihong@llu.edu (L.J.H.); mmomeni@llu.edu (M.M.); 2Department of Obstetrics & Gynecology, Southern California Permanente Medical Group, Fontana, CA 92335, USA; Sandy.x.huynh@kp.org; 3Department of Obstetrics & Gynecology, Division of Gynecologic Oncology, Loma Linda University Health, Loma Linda, CA 92354, USA; joykim@llu.edu; 4Department of Pathology, Loma Linda University School of Medicine, Loma Linda, CA 92354, USA; ldenham@llu.edu

**Keywords:** adenocarcinoma in situ, cervical cancer, fertility sparing, excisional margin status

## Abstract

**Background:** Adenocarcinoma in situ (AIS) of the cervix, is increasing in incidence, particularly in women of reproductive age. Fertility preservation is often desired. In a predominantly Hispanic population, we sought to determine the incidence of occult cervical cancer co-existing with AIS, and evaluate how conization margin status correlates with residual disease upon hysterectomy. **Methods:** A retrospective study utilizing a comprehensive cancer center database was conducted. Data from patients with histologically proven AIS of the cervix were abstracted. **Results:** Of 47 patients that met the criteria, 23 (49%) were Hispanic, 21 (45%) were White, two (4%) were Asian, and one (2%) was Black. The median age was 37. Forty-two patients underwent cervical conizations; 13/42 (48%) had positive margins upon conization; 28/42 (67%) underwent hysterectomies. Furthermore, 6/13 (46%) patients with positive conization margins had residual disease in hysterectomy specimens, with 2/13 (15%) found to have invasive cancer. In contrast, 0/14 (0%) of patients with negative margins had residual disease (*p* = 0.036, Chi-squared 4.41, df = 1). In total, 2/27 (7%) patients who underwent hysterectomies had invasive cancer (7%). **Conclusions:** Positive margins upon cervical conization for AIS of the cervix were correlated with a relatively high rate of residual AIS and occult invasive cancer. Negative conization margins were correlated with no residual disease. Those patients may be candidates for fertility-sparing treatment.

## 1. Introduction

Human papillomavirus (HPV) is the most common sexually transmitted infection in the United States of America (USA) [1]. Persistent infection with high-risk (HR) HPV is a necessary and causal catalyst for the development of almost all cases of cervical cancers [2,3]. Although most cervical cancer cases are of squamous histology (approximately 75%), 20–25% are of adenocarcinoma histology [4,5].

Adenocarcinoma in situ (AIS) of the uterine cervix, often detected incidentally upon cervical excisional biopsies, is the only known precursor of invasive adenocarcinoma of the cervix [6,7]. Although the incidence of squamous cell cervical carcinoma is declining, the incidence of AIS has increased dramatically over the last 50 years [8,9]. Historically, the finding of glandular lesion on a Papanicolaou test was a rare instance, occurring in less than 0.5% of tests [10]. Multiple factors may have contributed to this rise in the incidence of AIS, including better recognition by pathologists, but also the rise in HPV and in the incidence of these lesions [11,12,13]. The average age at diagnosis is 35–37 [13,14,15].

As recently ascertained by the HPV monitoring project, the incidence of AIS appears to be increasing in women in the 4th decade of life, and is stable in women aged 25–29 [14,15,16]. Early detection and appropriate management can prevent the occurrence of invasive disease, as the time for progression from AIS to invasive adenocarcinoma may be as long as 13 years, and is at least 5 years on average [13,17].

According to the recent Society for Gynecologic Oncology (SGO) evidence-based guidelines and recommendations, as well as the American Society for Colposcopy and Cervical Pathology (ASCCP) guidelines, hysterectomy is the preferred treatment of AIS post-excisional procedures [15,18]. However, fertility preservation is often desired for young women diagnosed with AIS. A desire for future fertility or a desire for conservative treatment (i.e., less extirpative surgery) has stemmed from a number of investigations into the safety of conization in lieu of hysterectomy. Unfortunately, no phase III trials exist to provide guidance in this situation; the best available data are retrospective. A number of investigations have examined factors associated with the safety of conization as a treatment of AIS, as opposed to simple or modified radical/radical hysterectomy. Data are limited by small sample sizes and a lack of long-term follow up. Negative margins upon conization appear to be the best predictive factor for residual disease and decreased odds of the recurrence of AIS, at 3%–12%, in variable populations [11,19,20,21,22,23,24,25,26,27,28,29]. A recent population-based Australian study demonstrated that after undergoing excisional treatment, the diagnosis of AIS was associated with a higher risk of persistence or recurrence, when compared to mixed AIS/CIN2/3 [30], inferring that fertility preservation should be carried out with caution.

In the current investigation, we aimed to gain a better understanding of the clinical behavior of AIS in the cohort of patients in San Bernardino and Riverside counties, the largest and 4th-largest counties in the US. Moreover, San Bernardino county has a large Hispanic population (54% of the population) [31], a patient group that is yet to be thoroughly studied in terms of the conservative management of AIS. The specific aim of this retrospective study was to determine how the presence of positive margins relates to residual AIS/invasive adenocarcinoma upon hysterectomy conducted for the definitive treatment of AIS post-excisional biopsies in our patient population. Secondarily, we aimed to investigate the incidence of occult cancer, coexisting with known adenocarcinoma in situ upon definitive treatment with hysterectomy or a re-excisional procedure for AIS. We also aimed to quantitate the rates of recurrence of AIS post-hysterectomy and re-excision for AIS.

## 2. Materials and Methods

This was a retrospective investigator-initiated study. After obtaining institutional IRB approval, the data registry from Loma Linda University Comprehensive Cancer Center (LLU-CCC), which is located in the San Bernardino County of Southern California, were queried for patients with the diagnosis of AIS of the uterine cervix. All patients over the age of 18 with histologically proven AIS of the uterine cervix as a diagnosis upon referral to gynecologic oncology were included in this retrospective chart review. The reviewed cases were patients that had been seen in the gynecologic oncology division by one of five attending gynecologic oncologists. All patients were seen between the period from 1 January 2018 up to 1 January 2018. The electronic medical record database Epic was used to abstract the data.

The following information was abstracted from the electronic medical record: histopathology, HPV status, demographics, parity, treatments, and follow-up information. Many of the original pathology reports did not comment on the distances of margins. Therefore, a secondary centralized histopathologic review was performed to confirm margin status and to document the distance to margins, carried out by a gynecologic pathologist (LD).

The margin status during excisional procedures was correlated with residual disease on hysterectomy. Furthermore, the patients with positive margins noted during excisional procedures were followed for evidence of recurrence. The statistical analysis was performed with GraphPad Prism (San Diego, CA, USA).

## 3. Results

Two hundred seventy-nine patient charts were identified via ICD diagnoses from the comprehensive cancer center tumor registry for preliminary review. Adenocarcinoma in situ of the uterine cervix as the presenting diagnosis for treatment via gynecologic oncology was used as the selection criteria for chart identification and abstraction. Of the 279 patients that were queried for adenocarcinoma in situ, 47 met the criteria for adenocarcinoma in situ of the uterine cervix based on a review of the available pathology reports confirming the above diagnosis.

### 3.1. Demographics

Of the 47 patients identified, the age at presentation ranged from 23 to 71 years. The median age at diagnosis of AIS for this group was 37, with the majority of patients aged from 23–44 (70%). By ethnic background: 23 of 47 (49%) were Hispanic, 21 of 47 (45%) were non-Hispanic White, two of 47 (4%) were Asian, and one of 47 was Black (2%).

In terms of parity, nine of 47 (19%) were nulliparous at the time of diagnosis, and at least nine patients desired future fertility preservation. The majority of patients, i.e., 37 of 47 (79%), were parous. Nine patients (19%) had one birth, 17 of 47 (36%) had two births, and 11 (23%) had three or more children.

Out of 47 patients presenting with AIS, 49% had known high-risk HPV (HR HPV) status upon presentation—21 (45%) were HR HPV-positive, two (4%) were negative, 51% were HR HPV status unknown upon presentation. A summary of the demographics is presented in Table 1.

### 3.2. Concurrent Invasive Adenocarcinoma upon Initial Presentation

Out of the 47 patients that met the criteria for adenocarcinoma in situ upon presentation, one patient (2%), had concurrent invasive adenocarcinoma at the time of the excisional procedure. The tumor was 3 cm in size with 9 mm of invasion, positive endocervical margins, and a 3 mm distance from the ectocervical margin. Lympho-vascular invasion was also present. This patient was diagnosed with stage IB_2_(r) disease and initiated chemoradiation. She had a complete response and no evidence of recurrence at the 1 year follow-up mark.

### 3.3. Positive Margins and Residual AIS on Subsequent Hysterectomy

Of the 47 that received the initial diagnosis of AIS, 42 patients (89%) went on to have a loop electrosurgical excisional procedure (LEEP) or cold knife conization (CKC). Twelve were LEEPs and 30 were CKCs. The other five patients (11%) went from cervical biopsy to definitive hysterectomy (Figure 1). One patient refused a cone biopsy after extensive counseling and desired a definitive hysterectomy. Two patients underwent surgery at outside facilities: one patient had abnormal uterine bleeding and underwent urgent supracervical hysterectomy, and the second was not offered LEEP or CKC for unknown reasons. The other two patients who did not undergo conization were diagnosed by outside institutions with EIN versus endometrioid adenocarcinoma of the uterus prior to undergoing extrafascial hysterectomies. Two of the five above-described patients ultimately had invasive adenocarcinoma on final pathology. These five patients were excluded from the final statistical analysis as they did not have undergo excisional procedures prior to their hysterectomies.

Of the 42 patients that had a diagnostic excisional procedure, 28 patients (66%) went on to have a subsequent hysterectomy. Twenty-seven of 28 (96%) of the post-conization patients had pathology slides available for institutional pathology review by LD. Of these 27, 13 patients had positive conization margins (48%), whether on LEEP or cone, and 14 had negative margins (52%).

Five out of the 13 patients (38%) with positive margins went on to have residual adenocarcinoma in situ on the hysterectomy specimen, with one out of those five (8%) having both invasive adenocarcinoma and AIS. Interestingly, one patient out of 13 (8%) had invasive cancer but no residual AIS on hysterectomy. In total, two of 13 patients (15%) with positive margins on conization were found to have invasive cancer on hysterectomy. Both patients were diagnosed with stage IB_1_(p) adenocarcinoma of the endocervix. One was lost to follow-up immediately after surgery. The other patient was followed for 1.5 years postoperatively and did not require any additional treatment during that follow-up period.

On the other hand, 0/14 (0%) of the patients with negative margins had residual AIS or cancer in the hysterectomy specimens (Table 2, Figure 2). Upon Chi-squared testing for trends, patients with negative conization margins had a significantly lower chance of having residual AIS and/or invasive cancer found in hysterectomy specimens, (*p* = 0.036, Chi-squared 4.41, df = 1).

### 3.4. Positive Margins and Post Excision Recurrence

Of the 42 patients that had diagnostic excisional procedures, 17 had positive margins (40%). Of these 17 patients with positive margins, 13 underwent subsequent hysterectomies (76%). Five out of the 13 (38%) hysterectomies had residual adenocarcinoma in situ, with one patient having invasive adenocarcinoma in addition to AIS (8%).

The one patient who had positive margins and invasive adenocarcinoma but no AIS on the hysterectomy specimen remained disease-free after 1.5 years but was lost to follow up thereafter. Unfortunately, three out of five (60%) patients with residual adenocarcinoma in situ, including the one with invasive cancer, were lost to follow up. Only two patients had postoperative Pap smears within 6 months, but they both were negative for recurrence. Unfortunately, most patients were lost to follow-up after 1.5 years (median length of follow up: 1 year), so it is unclear if patients ended up with recurrence further in the follow-up process.

For the eight patients who had positive margins on conization and had hysterectomies with no residual adenocarcinoma in situ at the time of hysterectomy, only two (25%) had abnormal Pap smears postoperatively. Five of the eight (62.5%) had normal pap smears postoperatively, and one (12.5%) was lost to follow-up.

For the four patients who had positive margins but did not undergo hysterectomies, two of the four underwent re-conizations, achieving negative margins. Of those two patients with re-conization, one moved away prior to follow up; the other had abnormal initial colposcopic biopsy but was not diagnosed with AIS recurrence at 1 year, when she had to switch providers for insurance reasons.

One of the two patients that did not undergo a re-excision had an abnormal Pap test at 6 months and then was lost to follow up; the other patient without re-excision has had normal cytology for 5 years.

## 4. Discussion

In this investigation, we reviewed all consecutive cases of histologically proven AIS presenting to a single-institution gynecologic oncology practice with a large fraction of Hispanic patients among the total patient population. In this study, we calculated the incidence of concurrent occult cancer at the time of definitive hysterectomy and the prognostic value of positive margins on the presence of residual AIS and/or occult cancer. In our study population of 47, with 28 patients undergoing hysterectomy, the incidence of invasive cancer for evaluable patients undergoing hysterectomy was 2/27, (7.4%). Another 2% (one of 47) had concurrent invasive carcinoma at the time of initial excision. A large retrospective study from the MD Anderson Cancer Center examined the outcomes of 180 patients with AIS; 70 patients ultimately underwent hysterectomies, 11% had residual AIS on hysterectomy specimens, and 2.8% showed invasive cancer on residual specimens. Out of 52 patients with negative conization margins who underwent hysterectomy, 6/52 (11.5%) had residual AIS, and 1/52 (1.9%) had invasive adenocarcinoma [23]. Three retrospective studies identified in the literature [12,32,33] followed women with positive and negative margins on conization, and found a fairly high rate of residual AIS on hysterectomy specimens even in patients with negative margins (29%, 45%, and 44%, respectively without available data on the incidence of invasive adenocarcinoma upon hysterectomy). Several other studies showed lower rates of residual AIS on hysterectomies with negative margins, i.e., 1/13 (8%) AIS, with 1/13 (8%) patients developing recurrent adenocarcinoma [27] and 6% AIS, 0% invasive carcinoma [11]. A study by Tierney and colleagues showed a 14% residual AIS rate, and a 0% invasive adenocarcinoma rate in patients with negative margins and negative ECC, although some of those procedures were re-conizations [26]. Another relatively older study, published in 1998, reported that only a negative endocervical margin of >10 mm had a reliable association with no residual AIS; otherwise, a negative endocervical margin was not a reliable predictor of no residual disease upon hysterectomy [34]. Given our limited study population and findings listed above, it is difficult to determine how our rate of residual AIS and invasive carcinoma would reliably translate to a larger, diverse general patient population. The findings of our study showed a similar significant occult invasive cancer rate upon hysterectomy post-excision. This finding does support current management guidelines, which include definitive hysterectomy as the standard treatment if patients have completed childbearing, as the rate of 7.4% of occult carcinoma detected via hysterectomy and not via the excisional procedure is rather high. As with all premalignant disease, early identification and surgical excision correlates with better overall survival.

We also sought to determine how specifically the presence of positive margins relates to residual adenocarcinoma in situ and/or invasive adenocarcinoma upon definitive hysterectomy, and conversely, if negative margins were associated with no residual disease. Our study indicated that positive margins indeed correlated with a high rate of residual AIS on the hysterectomy specimens, as 38% of the cases had subsequent residual disease. Similarly, negative margins were correlated with no residual AIS in final hysterectomy specimens. These findings were statistically significant and are important for the counseling and management of younger patients who are seeking fertility preservation. The median age of diagnosis of AIS of the cervix is in the mid-30s [13,14,15], with women frequently desiring fertility, with at least 9/47 (19%) desiring fertility in our cohort. Our study adds to a number of retrospective studies in the literature showing a low but not negligible rate of residual AIS in patients with negative margins on preceding conizations [11,23,26,35]. Our study had a rare finding of 0% residual AIS in patients with negative conization margins. Our study, although being relatively small, utilized hysterectomy specimens as a gold standard of diagnosis of residual disease. Although the SGO and ASCCP management guidelines do allow for the conservative management of patients with AIS desiring fertility with negative conization margins, patients should be counseled that the risk of residual AIS and even invasive adenocarcinoma is fairly low but is not negligible. The desire for fertility should be carefully weighed against the chance of occult invasive disease, as well as disease progression and persistence.

In our study, 49% of patients were of Hispanic/Latina decent, making it one of the very few studies in the literature examining a cohort with a large fraction of Hispanic women. A 2011 study by Tierney and colleagues described a cohort that was 73% Hispanic [26].

Finally, we attempted to quantify the rates of recurrence of adenocarcinoma in situ and or adenocarcinoma of the cervix post-excision and/or hysterectomy. This proved to be much more challenging than anticipated. Many of the patients were lost to follow-up after only a short period of time (approximately 1–1.5 years). This makes it difficult to determine how positive margins on initial excision ultimately affect overall prognosis and survival.

Other limitations to this study include a short study period and follow-up period and the relatively small sample size. Moreover, this was a single-institutional experience. Given the relatively recent transition to Epic as the primary electronic medical record system, our study period was limited to 6 years. The aforementioned factors limit the power of our investigation and the ability to generalize the findings to a larger population.

Future directions would include involving other comprehensive cancer centers, lengthening our study period, and improving long-term follow-up. In summary, our study shows that patients with positive margins would benefit from definitive treatment, i.e., hysterectomy, given the high incidence of residual disease, which portends a higher risk of recurrent disease or invasive cancer. Though our long-term follow-up was incomplete, additional surgical management would likely confer improved survival for patients diagnosed with AIS of the uterine cervix.

## 5. Conclusions

Although AIS of the uterine cervix is a premalignant disease, definitive hysterectomy would be recommended for all patients that have completed childbearing due to the significant risk of concurrent invasive carcinoma or residual AIS. Lastly, we prognosticate that for younger patients who are seeking fertility perseveration that negative margins correlate with a better chance of success with conservative management. This may be particularly applicable to a population with a high Hispanic/Latina contingency.

## Figures and Tables

**Figure 1 diagnostics-11-01889-f001:**
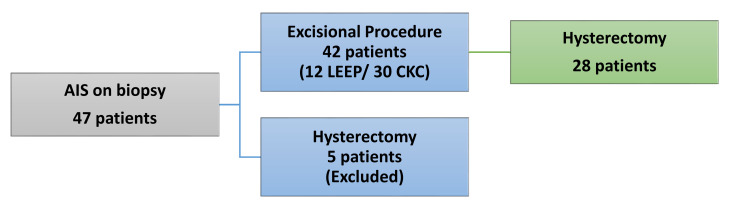
Flow diagram of patients who underwent excisional procedures followed by definitive hysterectomy.

**Figure 2 diagnostics-11-01889-f002:**
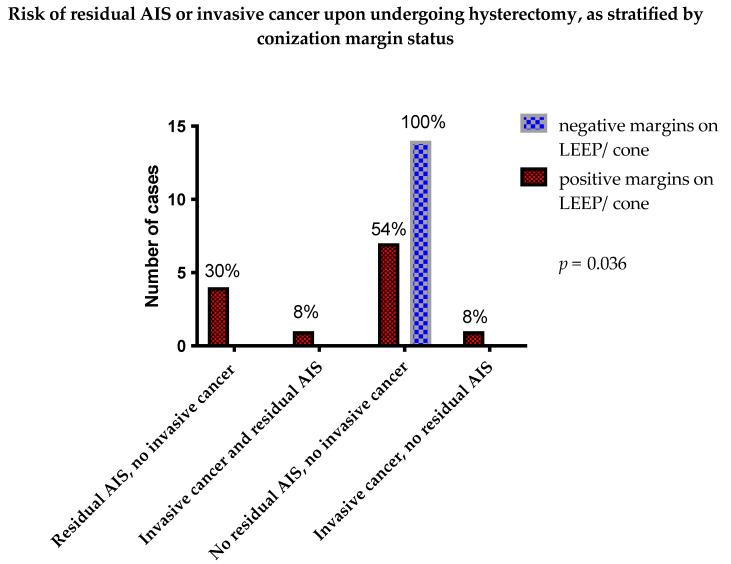
Presence of residual AIS or invasive cervical cancer in hysterectomy specimens, as stratified by conization margin status. Patients with negative conization margins had a significantly lower chance of having residual AIS and/or invasive cancer found in hysterectomy specimens (0% vs 46%) when analyzed with Chi-squared test, *p* = 0.036, Chi-square 4.42, df = 1).

**Table 1 diagnostics-11-01889-t001:** Cohort demographics: age, race, high-risk HPV status and parity upon presentation for treatment.

Age (Years) Median: 37 (23–71)		Race (*n* = 47)		High Risk HPV Status (*n* = 47)		Parity (*n* = 46)	
23–34	19 (40%)	Hispanic	23 (49%)	Positive	21 (45%)	0	9 (19%)
35–44	14 (30%)	White	21 (45%)	Negative	2 (4%)	1	9 (19%)
45–64	12 (26%)	Asian	2 (4%)	Unknown	24 (51%)	2	17 (36%)
65+	2 (4%)	Black	1 (2%)			3+	11 (23%)

**Table 2 diagnostics-11-01889-t002:** Specimen margin status upon cervical conization as correlated with findings of residual disease on post-conization hysterectomy specimens.

Hysterectomy Specimens	Conization Specimens		Total
	Positive Margins on LEEP/Cone	Negative Margins on LEEP/Cone	
Residual AIS	4 (30%)	0 (0%)	
Residual AIS and residual invasive cancer	1 (8%)	0 (0%)	5 (19%)
No residual AIS	7 (54%)	14 (100%)	
Invasive cancer, no residual AIS	1 (8%)	0 (0%)	22 (81%)
Total	13 (100%)	14 (100%)	27 (100%)

## Data Availability

Data for review will be furnished upon reasonable request.

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
