# Peer review of "Margin Status Post Cervical Conization Predicts Residual Adenocarcinoma In Situ (AIS) and Occult Adenocarcinoma in a Predominantly Hispanic Population"

_diagnostics, 2021, doi:10.3390/diagnostics11101889_

Round 1

Reviewer 1 Report

The paper by Hong et al. entitled "Margin status post cervical conization predicts residual adenocarcinoma in situ (AIS) and occult adenocarcinoma in a predominantly Hispanic population" reports the retrospective analysis of a series of 27 patients with AIS diagnosed in LEEP/cone specimens that underwent hysterectomy with histologic sections available for the study. The analysis compared margin status on LEEP/cone specimens and histology on hysterectomy specimens. Positive margins on LEEP/cone specimens were observed in 13/27 cases. In 6 of these 13 cases (46%) residual disease was found in the hysterectomy specimen, corresponding to AIS (4 cases) or to invasive adenocarcinoma (2 cases), with (1 case) or without (1 case) associated AIS. In contrast, none of the 14 cases with clear margins was found to be associated with residual disease on the hysterectomy (p=0.036).

The study is well conducted and the paper well written. In spite of the limited number of cases analyzed, the results are significant and, regarding the scarcity of the data available on this disease, the paper is informative and deserves publication.

Minor remarks.

1) It is difficult from the abstract to get at first glance what was found in the hysterectomy specimens according to the margin status on cones. This comes from the fact that the data "with 2/27 having invasive cancer" is provided before the result “6/13 patients with positive conization margines had residual disease in the hysterectomy specimens” and that 2 of these 6 cases correspond to the invasive carcinoma cases. A formulation like that proposed above might improve the abstarct.

2) The table 1 is difficult to read and largely redundant with the data provided in the text. It is tempting for the reader to search for any correspondance between data on lines and columns. Perpendicular instead of horizontal lines should be used to clearly separate the columns and the last line (median age, range , total n= 47) could be suppressed..

3) This study reports that 45% of the cases were associated with high-risk HPV genotypes. Are there any data on the respective genotypes identified in these AIS? It would be interesting to provide these data if they are easily available.

Author Response

  • It is difficult from the abstract to get at first glance what was found in the hysterectomy specimens according to the margin status on cones. This comes from the fact that the data "with 2/27 having invasive cancer" is provided before the result “6/13 patients with positive conization margines had residual disease in the hysterectomy specimens” and that 2 of these 6 cases correspond to the invasive carcinoma cases. A formulation like that proposed above might improve the abstarct.

We’d like to thank Reviewer #1 for this impactful comment. The Results section of the abstract was revised to reflect the proposed structure. Please see below and in the edited manuscript:

Abstract: Background: Adenocarcinoma in situ (AIS) of the cervix, is increasing in incidence, particularly in women of reproductive age. Fertility preservation is often desired. In a predominantly Hispanic population, we sought to determine the incidence of occult cervical cancer co-existing with AIS, and evaluate how conization margin status correlates with residual disease upon hysterectomy. Methods: A retrospective study utilizing a comprehensive cancer center database was conducted. Data from patients with histologically proven AIS of the cervix were abstracted. Results:  Of 47 patients that met criteria, 23 (49%) were Hispanic, 21 (45%) White, 2 (4%) Asian, and 1 (2%) Black. Median age was 37. 42 patients underwent cervical conizations; 13/42 (48%) had positive margins upon conization; 28/42 (67%) underwent hysterectomies. 6/13 (46%) patients with positive conization margins had residual disease in hysterectomy specimens, with 2/13 (15%) found to have invasive cancer. In contrast, 0/14 (0%) of patients with negative margins had residual disease (p = 0.036, Chi-square 4.41, df = 1). In total, 2/27 (7%) patients who underwent hysterectomies had invasive cancer (7%). Conclusion:  Positive margins upon cervical conization for AIS of the cervix were correlated with a relatively high rate of residual AIS and occult invasive cancer. Negative conization margins were correlated with no residual disease. Those patients may be candidates for fertility sparing.

  • The table 1 is difficult to read and largely redundant with the data provided in the text. It is tempting for the reader to search for any correspondance between data on lines and columns. Perpendicular instead of horizontal lines should be used to clearly separate the columns and the last line (median age, range , total n= 47) could be suppressed..

Thank you for input, we have condensed the table (as below). We do think it is useful as a visual summary of our patient cohort, and, if possible, would like to leave it as part of the manuscript.

Age (years)

Median: 37 (23-71)

Race

(n=47)

High risk HPV status

(n=47)

Parity

(n=46)

23 – 34     19 (40%)

Hispanic        23 (49%)

Positive        21 (45%)

0             9 (19%)

35 – 44     14 (30%)

White             21 (45%)

Negative        2 (4%)

1             9 (19%)

45 – 64     12 (26%)

Asian               2 (4%)

Unknown     24 (51%)

2           17 (36%)

65+             2 (4%)

Black               1 (2%)

3+         11 (23%)

 Table 1. Cohort demographics: age, race, high risk HPV status and parity upon presentation for treatment.

3) This study reports that 45% of the cases were associated with high-risk HPV genotypes. Are there any data on the respective genotypes identified in these AIS? It would be interesting to provide these data if they are easily available.

Thank you for this insightful comment. We wish we could provide more meaningful data on this subject. After reviewing our data spreadsheet, 20/21 high risk HPV positive patients did not HPV genotype specified. 1/21 high risk HPV positive patients was specified to be HPV 18 positive. In light of paucity of data, we did not make further revisions.

Reviewer 2 Report

This is a retrospective study with small number of patients with AIS.

There are no informative data and new findings. 

Author Response

We do agree with the Reviewer that this is a relatively small, retrospective study, exploring the subject that has previously been approached in the literature.

With all due respect, we do, however, believe our study provides further insight into a controversial subject (i.e. hysterectomy vs conservative treatment of AIS). Extensive review of available literature conducted over the course of this project indicated a variable and often high rate of residual AIS and even cancer upon hysterectomy conducted post cervical conization. Our study does offer further insight into this matter by stratifying margin status as a strong predictor of no residual disease. In addition, our study population is predominantly Hispanic. There is paucity of data regarding outcomes in this population, and this in itself, is a contribution to the literature.

Reviewer 3 Report

In the present research, the authors show that margin status post cervical conization predicts residual adenocarcinoma in situ (AIS) and occult adenocarcinoma in a predominantly Hispanic population. The manuscript has novelty and the authors contribute in clinical practice. Those patients, with negative conization margins, may be candidates for fertility sparing.

Author Response

We appreciate the positive review and the elegant summary of our findings. There were no points to revise for this reviewer.

Reviewer 4 Report

The manuscript describes a retrospective evaluation of AIS recurrence following conization treatment. The manuscript is well written and objective.

Two minor comments/ questions below,

What was the medical decision to move on to hysterectomy on those 14 patients that had negative margins and no post residual AIS?

The authors focused on providing insight on research into the Hispanic population, were there differences between hispanic and white women?

Author Response

The manuscript describes a retrospective evaluation of AIS recurrence following conization treatment. The manuscript is well written and objective.

Two minor comments/ questions below,

What was the medical decision to move on to hysterectomy on those 14 patients that had negative margins and no post residual AIS?

The patients did not wish to preserve fertility; they were offered standard of care treatment in the form of hysterectomy for definitive management of AIS. As there are no prospective trials, we only have the data from retrospective trials with regards to residual disease post conization with positive or negative margins. The data varies in terms of the rate of residual disease on hysterectomy specimens, even with negative margins. Therefore, leaving the uterus in situ in patients not desiring future fertility may not be an entirely safe option.  

The authors focused on providing insight on research into the Hispanic population, were there differences between hispanic and white women?

In this study, we aimed to describe the pattern of residual AIS post conization in a predominantly Hispanic population to add to the literature with paucity of data on this subject. Our study did not examine the differences between subpopulations. Our data numbers are too small for that purpose.

Round 2

Reviewer 2 Report

i think this manuscript did not showed any new findings, with small number of patient.